# How seasons, weather, and part of day influence baseline affective valence in laboratory research participants?

**Maciej Behnke**[1]*, **Hannah Overbye**[2], **Magdalena Pietruch**[1], **Lukasz D. Kaczmarek**[1]

1 Faculty of Psychology and Cognitive Science, Adam Mickiewicz University, Poznań, Wielkopolska, Poland,
2 Department of Communication, UC Santa Barbara, Santa Barbara, California, United States of America

* macbeh@amu.edu.pl

## Abstract

Many people believe that weather influences their emotional state. Along similar lines, some researchers in affective science are concerned whether testing individuals at a different time of year, a different part of the day, or in different weather conditions (e.g., in a cold and rainy morning vs. a hot evening) influences how research participants feel upon entering a study; thus inflating the measurement error. Few studies have investigated the link between baseline affective levels and the research context, such as seasonal and daily weather fluctuation in temperature, air pressure, and sunshine duration. We examined whether individuals felt more positive or negative upon entering a study by clustering data across seven laboratory experiments (total $N$ = 1108), three seasons, and daily times ranging from 9 AM to 7 PM. We accounted for ambient temperature, air pressure, humidity, cloud cover, precipitation, wind speed, and sunshine duration. We found that only ambient temperature was a significant predictor of valence. Individuals felt more positive valence on days when it was cooler outside. However, the effect was psychologically negligible with differences between participants above c.a. 30 degrees Celsius in ambient temperature needed to generate a difference in affective valence surpassing one standard deviation. Our findings have methodological implications for studying emotions by suggesting that seasons and part of the day do not matter for baseline affective valence reported by participants, and the effects of ambient temperature are unlikely to influence most research.

## Introduction

Humans engage in daily activities that elicit positive and negative affect. For instance, people perceive favorable activities—going on a trip with friends and playing in the park with the child, lying in a hammock overlooking the beach—as eliciting positive affect [1]. On the other hand, people evaluate unfavorable activities—spending the holidays alone, having a home destroyed by a tornado, being struck by lightning—as eliciting negative affect [1]. These activities are often determined by contextual factors such as weather conditions and time cycles, including seasons and days. Although the activities themselves mostly determine the emotional

**Funding:** Preparation of this article was supported by National Science Center (Poland) research grants (UMO-2014/15/B/HS6/02418 to LDK; UMO-2014-15/N/HS6/04151 and UMO-2015/17/N/HS6/02794 to other members of Psychophysiological Laboratory at Faculty of Psychology and Cognitive Science at Adam Mickiewicz University) and Polish Ministry of Science and Higher Education Diamond Grant (DI2014011344, to another member of Psychophysiological Laboratory at Faculty of Psychology and Cognitive Science at Adam Mickiewicz University), and doctoral scholarship from National Science Centre in Poland (UMO-2019/32/T/HS6/00039) to MB.

**Competing interests:** The authors have declared that no competing interests exist.

experience, the contextual factors also impact how positive or negative people feel in the present moment [2]. Thus, in this study, we focused on contextual factors that are believed to influence emotional affect, namely weather, seasons, and parts of the day.

As for seasons, people feel the worst in winter and feel the best in summer [3, 4]. For instance, one seasonality's negative effect on affect is conceptualized as winter seasonal affective disorder [5]. The seasonal affective disorder is described in DSM as a variant of categorical mood disorder. However, it has been clear that the tendency of people to experience seasonal changes in mood and behavior is not limited to those severely affected but has an impact on the normal population as well [6–8]. The review of epidemiological research on seasonal affective disorder found that almost in all studies, seasonal variations in mood were found with the depressive symptoms usually peaking in winter [7]. Although most studies used self-report measures in which individuals reported when they felt best or worst during the past year, others measured the mood successively across seasons and supported the mood drops in winter [7].

Furthermore, as almost all processes with a physiological component [9], affect–especially positive affect—also has diurnal cycle components [10–13]. Individuals have an endogenous circadian system that operates with solar time in the day-night cycle [14, 15]. However, there is no consensus related to the peak of affective experience during the day, with research showing conflicting information for when people feel best throughout the day [11, 13, 16]. Studies have shown that people feel the best in the morning [17, 18] around 11:00 [19], around the middle of the day [12, 20], in the middle of the afternoon [11], and in the evenings [13, 16].

Affect is also influenced by weather conditions [21, 22]. For instance, individuals tend to feel better when the day is less cloudy [23–26], barometric pressure is higher [27], precipitation lower [24, 28], and wind power stronger [23]. However, more recent large-scale studies have not replicated many previous findings regarding the link between weather and emotions [29, 30]. For instance, contradictory effects have primarily been focused on temperature. Some researchers have found that temperature increases are associated with increased positive affect and reduced negative affect [24, 27, 28, 31], whereas others have found the opposite [23, 32–34].

The contradictory results may stem from not accounting for seasons and parts of the day when studying weather and affect associations. For instance, the negative association between the daily variation of affect and temperature or sunshine may be driven by the fact that people are more positive at night simply because they are not working, without any relation to lower temperature and sunshine [34]. Furthermore, affect fluctuations related to weather conditions within a single season may not translate into the between-seasons scale. Studies have shown that affect is positively associated with air temperature and barometric pressure in spring [27] but negatively associated in summer [32]. Thus, in our study, we investigated whether people feel differently across time cycles (e.g., seasons) and whether the weather conditions might explain these effects.

Furthermore, the differences between the findings may also stem from differences in used methodologies. For instance, to measure participants' mood studies used single-item questions [24, 25, 29, 30], standardized questionnaires [23, 27], and automatic mood detection from social media posts [16, 31]. Some studies used a single report from participants [25, 27, 30], whereas other employed repeated measure design with daily diaries [2, 20, 23, 24, 29, 35], and experience sampling methods [36]. Furthermore, the repeated measures lasted from reports over 11 days [24]; 14 days [36], 25 days [37], 30 day [35]; 90 days [2, 20], to two-year period [23]. Thus, the variety of methods may be related to the inconsistencies of the results.

Examination of how individuals feel depending on weather or point in the daily or yearly cycle is also essential for the perspective of advancing laboratory research methodology. Among concerns of experimental researchers is whether testing participants on different

occasions significantly increases the noise in the measurement or produces a systematic bias [38]. This is a special case of a more general methodological problem of whether research results are influenced by contextual (and seemingly trivial) factors, e.g., a time point in the semester among students [39–41]. Thus, we focused on baseline affect among laboratory research participants. Namely, we aimed to examine whether individuals transfer any affective results of weather and time cycle into the research. If this was the case, testing individuals in more restricted conditions (e.g., at a similar part of a day or halting data collection in case of significant weather change) might improve the data quality and–possibly—reduce the type II error. If this was not the case, ignoring weather effects or time cycles might decrease the planning burden on the researcher, improve the data collection flow, and extend the time available for data collection in a project.

There is a meaningful difference between the practical methodological aim of our approach and studies that asked about the general link between weather and emotions. Most studies have been concerned with individuals' diverse daily settings. In contrast, we focused on how individuals feel in a well-controlled laboratory environment that isolates the room environment from the outdoor environment. For instance, most researchers follow the recommendations and aim to maintain stable room temperature within the range of thermal comfort, set the constant light intensity and color temperature by covering windows and using artificial light, and reduce environmental noises (e.g., rain or wind) via the room's sound attenuation [38, 42–44]. Consequently, this might reduce the impact of weather on affect in this specific group. Moreover, laboratory research often focuses individuals on upcoming tasks. Thus participants might be temporarily less aware of daily factors such as weather, part of day, or seasonal activities. This might even further reduce the impact of weather or time cycles on participants' affect.

### Present study

We used the novel approach to examine the time-rhythmic and weather characteristics of affective experience among laboratory research participants. We focused on affective valence, which is the most fundamental aspect of humans' emotional response [45]. The reported affect was recorded as a part of the psychophysiological baseline and reflected the resting state before the beginning of the main experiment. The data for this investigation were collected over four years (from November 2016 to March 2019) from seven different laboratory experiments in a mild continental climate in Central Europe.

The uniqueness of our approach is threefold. First, all participants were tested in the same laboratory conditions, without the possible confounding influences of affect-associated behaviors that might bias reported affect in studies using diary methodology [23, 24], experience sampling design [36], or data collected in population studies [30]. Second, we used precise weather conditions that occurred during the experiments rather than using weather variables for the day of self-reports [30, 32]. Third, unlike other studies that examined the association between weather and affect during a single season [29, 32, 36], we investigated data collected during three seasons–winter, spring, and autumn.

## Materials and methods

The data for this study were derived from seven laboratory experiments that examined the psychophysiology of emotions [46–50]. Details about the studies are presented in (S1 Table).

### Participants

We collected data from 1108 individuals (47% female) that were tested in the same laboratory in Poznan, Poland. Participants were in the age between 18 and 38 ($M = 21.86$; $SD = 2.65$). All

participants were Caucasian. A power analysis using G\*Power 3.1 [51] indicated that examining 954 participants would allow us to detect small effect sizes of $f^2 = 0.02$, with the power of 0.80, for the regression coefficient. Before participating in each study, we asked volunteers to reschedule if they experienced illness or a major negative life event to eliminate factors that might influence the emotional experience. Each participant provided written informed consent and received vouchers for a cinema ticket for participation in the study. The Institutional Ethics Committee at the Faculty of Psychology and Cognitive Science, Adam Mickiewicz University, approved all seven studies.

## Measures

**Emotional valence.** Participants continuously reported how they felt using a Response Meter (ADInstruments, New Zealand) with a scale ranging from 1 ("extremely negative") to 10 ("extremely positive"). Above the numeric scale, we provided a negative-positive valence graphical scale modeled after the self-assessment manikin [52]. A similar approach was employed in previous studies of the impact of time rhythms and weather of affect [13, 53]. The data was recorded with Powerlab and processed with LabChart 8.19 software (ADInstruments, New Zealand). Participants continuously reported their affect, while waiting for the five minutes without doing any unnecessary actions (resting baseline). We calculated the mean affective valence from the last two minutes of baseline to account for the part of the baseline that was the most proximal to the study and to limit the influence of interaction with the experimenter on affect. Electronic rating scales collect reliable and valid emotion ratings [54–56].

**Weather data.** We used weather data from the weather station in Poznan, collected by the Polish Institute of Meteorology and Water Management. The weather variables were matched to the experimental data of the participants by date and hour. We examined the impact of the following indicators: ambient temperature, air pressure, humidity, cloud cover, precipitation, wind speed, sunshine duration. Table 1 presents means and standard deviations for weather conditions per season, day of the week, and part of the day.

**Time rhythms.** We examined the impact of time rhythms on affect in two ways, i.e., including seasonal and daily variations. We clustered the data based collection moment:

**Table 1. Descriptive characteristics of affect and weather conditions.**

|  | N | Affect | | Temperature | | Air Pressure | | Humidity | | Cloud cover | | Precipitation | | Sunshine duration | | Wind speed | |
|---|---|---|---|---|---|---|---|---|---|---|---|---|---|---|---|---|---|
|  |  | *M* | *SD* | *M* | *SD* | *M* | *SD* | *M* | *SD* | *M* | *SD* | *M* | *SD* | *M* | *SD* | *M* | *SD* |
| Season |  |  |  |  |  |  |  |  |  |  |  |  |  |  |  |  |  |
| Winter | 520 | 5.03 | 0.91 | 3.21 | 5.16 | 1007.86 | 11.53 | 57.65 | 40.05 | 5.37 | 2.69 | 0.13 | 0.55 | 0.28 | 0.40 | 1.24 | 1.10 |
| Spring | 252 | 5.50 | 1.26 | 19.53 | 4.69 | 1005.77 | 6.56 | 49.10 | 21.18 | 3.99 | 2.65 | 0.18 | 0.77 | 0.70 | 0.42 | 1.53 | 1.43 |
| Autumn | 336 | 5.13 | 1.04 | 5.15 | 3.95 | 1006.41 | 9.41 | 49.18 | 43.10 | 6.79 | 4.37 | 0.24 | 0.75 | 0.13 | 0.29 | 1.45 | 1.35 |
| Part of the day |  |  |  |  |  |  |  |  |  |  |  |  |  |  |  |  |  |
| 9:00–11:00 | 68 | 5.32 | 1.26 | 9.62 | 9.10 | 1007.21 | 8.90 | 76.79 | 25.07 | 4.96 | 2.78 | 0.08 | 0.27 | 0.50 | 0.48 | 1.44 | 1.09 |
| 11:01–13:00 | 216 | 5.19 | 1.01 | 8.98 | 8.69 | 1007.64 | 10.01 | 48.17 | 37.39 | 5.41 | 2.60 | 0.14 | 0.68 | 0.50 | 0.46 | 1.56 | 1.40 |
| 13:01–15:00 | 260 | 5.18 | 1.10 | 8.71 | 8.29 | 1006.41 | 9.37 | 51.71 | 35.44 | 5.84 | 4.93 | 0.20 | 0.66 | 0.45 | 0.45 | 1.50 | 1.35 |
| 15:01–17:00 | 282 | 5.23 | 1.11 | 8.45 | 8.74 | 1007.00 | 9.86 | 50.44 | 36.71 | 5.23 | 2.77 | 0.21 | 0.83 | 0.35 | 0.43 | 1.24 | 1.11 |
| 17:01–19:00 | 282 | 5.14 | 1.01 | 7.10 | 8.05 | 1006.39 | 10.02 | 53.83 | 38.31 | 5.18 | 2.89 | 0.18 | 0.62 | 0.12 | 0.30 | 1.22 | 1.22 |

*Notes.* N = number of participants. Units: Affect = 0–10 Likert scale points, Temperature = Celsius degrees, Air Pressure = Millibar, Humidity = percentage of saturated air at 0 degrees Celsius, Cloud Cover = 0–8 Oktas, Precipitation = millimeters, Sunshine duration = percentage of sunshine during given hour, Wind Speed = meters per second.

season (winter, spring, autumn), and part of the day (early mornings, 9:00–11.00; late morning, 11:01–13:00, early afternoon, 13:01–15:00, late afternoon, 15:01–17:00, and early evening, 17:01–19:00). The number of laboratory visits per season, and part of the day are presented in Table 1.

## Analysis

**Preliminary analysis.** We examined whether the participants tested across seasons and parts of the day differed in age, sex, and BMI, using univariate analysis of variance. To examine differences between the seasons, we used post hoc tests with Bonferroni correction. To address multicollinearity between predictors of affect in our analysis we calculated variance inflation factor (VIF), with values < 5.00, and Tolerance with values > .20 indicating acceptable level of multicollinearity between variables [57, 58].

**Main analysis.** We examined the rhythmic characteristics of affective valence, including the impact of weather using two-level path analysis with maximum likelihood estimation with robust standard errors (MLR) in mPlus 8.0 [59, 60]. We regressed the affective valence on the mediators (weather conditions) and independent variables (seasons, part of the day). We controlled for age and sex by introducing it as a covariate for affective valence (Fig 1). We dummy-coded seasons and parts of the day, such that significant differences in the model accounted for differences relative to the winter and early mornings, respectively. In the two level-model, we nested individuals data within the studies. We calculated RMSEA, the recommended fit index for the MLR. RMSEA estimator with values < .08, along with the CFI with values above .90, indicates acceptable fit [61]. To interpret the strength of regression coefficients, we used standardized β as an indicator of 0.10 small, 0.30 medium, and 0.50 large effect sizes [62, 63].

## Results

### Preliminary analysis

We found that the samples examined across the seasons differed in participants age, $F$ (2, 1105) = 10.32, $p < .001$, sex, $F$ = (2, 1105) = 6.53, $p = .002$, but not BMI, $F$ (2, 1005) = 0.88, $p = .42$. The post-hoc tests showed that we tested more women in winter and autumn than in spring, $ps < .001$. Furthermore, we tested younger participants in spring than in winter and autumn, $ps < .05$. We found that the samples examined across the parts of the day differed in participants sex, $F$ (4, 1103) = 2.75, $p = .03$, but not age $F$ (4, 1103) = 0.17, $p = .95$, nor BMI, $F$ (4, 1003) = 1.70, $p = .15$. The post hoc tests did not show any difference between the groups. Based on these preliminary results we controlled for participants age and sex in our main

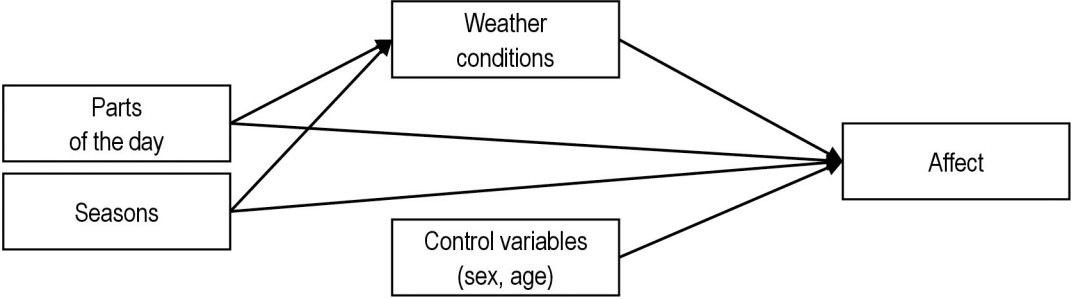

**Fig 1. Model for role of seasons, time of day, and weather conditions in affective valence.** For presentation simplicity, we grouped weather conditions and control variables in this figure. We examined paths for each variable separately.

analysis. We found that multicollinearity indices for all predictors of affects were in the recommended range VIFs < 4.09 and Tolerances > .24.

## Main analysis

Descriptive statistics and correlation between study variables are presented in Tables 1 and 2. The path model fit the data well, RMSEA = .04, 90% CI [.03, .05], CFI = .94 (Fig 2). For clarity, Fig 2 presents only significant paths. Table 3 presents detailed results.

**Role of seasons, part of the day, and weather conditions in affective valence.** We found a positive direct effect of spring on valence (Table 3) and a negative indirect effect of spring on valence via ambient temperature β = -.20, 95% CI [-.31, -.094]. These two opposing effects canceled each other out, producing a non-significant total effect of spring on valence, β = .21, 95% CI [-.12, .54]. This decomposition of the total effect suggests that participants would feel generally better in spring than in winter if not adverse effects of higher temperatures in spring. Yet, given their joint influence, the effect of spring on valence was non-significant. We found no difference in affective valence between parts of the day.

Of the weather conditions, only the ambient temperature predicted the participants' affective valence (Table 3). Participants felt better when it was cooler outside. The unstandardized estimate was b = -.03, showing that a decrease of one degree Celsius predicted an increase in an individual's affect by 0.03 points on the scale from one to ten, an equivalent of a 3.32% valence SD. To further support our findings, we run an exploratory analysis, in which we tested the model for each season separately. We found that people felt better when it was colder outside in spring β = -.27, 95% CI [-.40, -.14]. and in autumn β = -.12, 95% CI [-.24, -.01]. The relationship in winter was not significant β = -.005, 95% CI [-.09, .08].

**Seasons and weather conditions.** Relative to winter, the ambient temperature was higher in spring but not in autumn (Table 3). The wind speed was higher in spring and in autumn

**Table 2. Correlations among study variables.**

|  | M | SD | 1. | 2. | 3. | 4. | 5. | 6. | 7. | 8. |
|---|---|---|---|---|---|---|---|---|---|---|
| 1. Affect | 5.19 | 1.08 |  |  |  |  |  |  |  |  |
| 2. Temperature | 7.48 | 8.18 | .09** |  |  |  |  |  |  |  |
| 3. Air Pressure | 1006.91 | 10.06 | 0.01 | -.27** |  |  |  |  |  |  |
| 4. Humidity | 53.94 | 37.99 | 0.01 | -.22** | -.08** |  |  |  |  |  |
| 5. Cloud Cover | 5.53 | 3.38 | -.07* | -.24** | -.11** | .12** |  |  |  |  |
| 6. Precipitation | 0.18 | 0.71 | -0.01 | .06* | -.20** | .15** | .14** |  |  |  |
| 7. Sunshine | 0.33 | 0.43 | .06* | .49** | 0.05 | -.21** | -.49** | -.11** |  |  |
| 8. Wind speed | 1.37 | 1.26 | 0.04 | .10** | -0.06 | -.10** | 0.06 | -0.02 | .11** |  |
| 9. Late mornigns | 0.20 | 0.40 | 0.01 | 0.04 | 0.05 | -0.07* | 0.02 | -0.02 | 0.17** | 0.07* |
| 10. Early afternoon | 0.24 | 0.42 | -0.03 | 0.02 | -0.02 | -0.02 | 0.07* | 0.02 | 0.14** | 0.05 |
| 11. Late afternoon | 0.25 | 0.43 | 0.02 | -0.01 | 0.01 | -0.03 | -0.02 | 0.04 | -0.01 | -0.06 |
| 12. Early evening | 0.25 | 0.44 | -0.02 | -0.07* | -0.02 | 0.02 | -0.05 | -0.01 | -0.33** | -0.06* |
| 13. Spring | 0.26 | 0.42 | 0.20** | 0.81** | -0.08** | -0.04 | -0.22** | 0.04 | 0.48** | 0.06* |
| 14. Autumn | 0.30 | 0.46 | -0.03 | -0.19** | -0.03 | -0.10** | 0.25** | 0.05 | -0.31** | 0.05 |
| 15. Sex | 0.47 | 0.50 | -.09** | -.10** | 0.01 | -.01 | .03 | -.06 | .01 | -.02 |
| 16. Age | 21.86 | 2.65 | 0.01 | .11** | -0.01 | -0.04 | -0.04 | -0.04 | .09** | -.09** |

*Notes.* Sex coded as men = 0, and women = 1; Sunshine = Sunshine duration; Seasons dummy-coded relative to winter; Parts of day dummy-coded relative to early morning.

* p < .05

**p < .01.

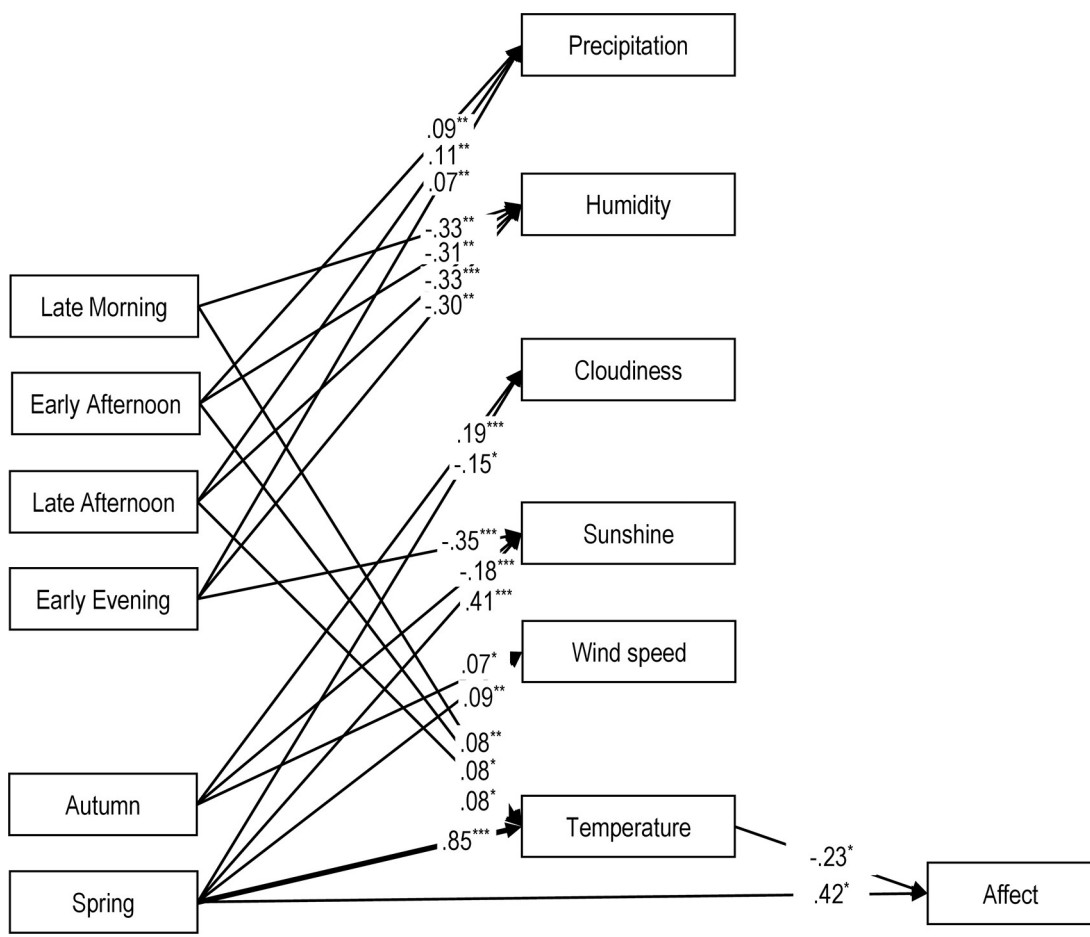

**Fig 2. Path model for role of seasons and weather conditions in affective valence.** *Note.* The figure presents only significant paths for the tested path model. For clarity, all non-significant paths from the model were omitted. Thicker lines represent stronger effects. Sex coded as 0 = men, 1 = women. *$p < .05$, **$p < .01$, ***$p < .001$.

than in winter. The cloudiness was higher, and the sunshine duration was lower in autumn when compared to winter. In contrast, the cloudiness was lower, and the sunshine duration was higher in spring when compared to winter. We found no differences between the seasons in precipitation, air pressure, and humidity (Table 3).

**Part of the day and weather conditions.** The early mornings were more humid than late mornings, early afternoons, late afternoons, and early evenings (Table 3). The precipitation was higher in early afternoons, late afternoons, and early evenings when compared to early mornings. The sunshine duration was shorter in the early evenings than in the early mornings. We found no differences in temperature, air pressure, cloudiness, and wind speed between the parts of the day (Table 3).

## Discussion

We examined whether individuals who start a laboratory experiment report different levels of affect depending on contextual factors such as season, part of a day, and weather conditions. We found that research participants felt better when it was colder outside. However, this effect had negligible practical meaning. Any differences in reported affect would vary within one standard deviation as long as the differences in the temperature between participants on

**Table 3. Path analysis details.**

| Outcome | Predictors | Std. Estimate | SE | p |
|---|---|---|---|---|
| Affect | | | | |
| | Temperature | -0.23 | 0.10 | 0.03 |
| | Sex | -0.06 | 0.05 | 0.23 |
| | Affect | -0.03 | 0.02 | 0.20 |
| | Cloudiness | -0.07 | 0.04 | 0.09 |
| | Sunshine | -0.07 | 0.05 | 0.13 |
| | Wind speed | 0.02 | 0.04 | 0.61 |
| | Precipitation | 0.00 | 0.04 | 0.93 |
| | Air pressure | -0.02 | 0.03 | 0.34 |
| | Humidity | -0.02 | 0.03 | 0.51 |
| | Spring | 0.42 | 0.21 | 0.04 |
| | Autumn | 0.07 | 0.05 | 0.17 |
| | Late Morning | 0.01 | 0.06 | 0.92 |
| | Early Afternoon | -0.03 | 0.07 | 0.71 |
| | Late Afternoon | 0.00 | 0.08 | 0.96 |
| | Early Evening | -0.06 | 0.08 | 0.44 |
| Temperature | | | | |
| | Spring | 0.85 | 0.12 | 0.00 |
| | Autumn | 0.12 | 0.16 | 0.47 |
| | Late Morning | 0.08 | 0.06 | 0.18 |
| | Early Afternoon | 0.08 | 0.05 | 0.10 |
| | Late Afternoon | 0.08 | 0.05 | 0.11 |
| | Early Evening | 0.03 | 0.06 | 0.60 |
| Air pressure | | | | |
| | Spring | -0.11 | 0.08 | 0.16 |
| | Autumn | -0.07 | 0.07 | 0.27 |
| | Late Morning | 0.03 | 0.02 | 0.13 |
| | Early Afternoon | -0.03 | 0.02 | 0.26 |
| | Late Afternoon | -0.01 | 0.02 | 0.70 |
| | Early Evening | -0.03 | 0.03 | 0.35 |
| Humidity | | | | |
| | Spring | -0.10 | 0.25 | 0.69 |
| | Autumn | -0.13 | 0.37 | 0.73 |
| | Late Morning | -0.33 | 0.11 | 0.00 |
| | Early Afternoon | -0.31 | 0.09 | 0.00 |
| | Late Afternoon | -0.33 | 0.09 | 0.00 |
| | Early Evening | -0.30 | 0.11 | 0.01 |
| Cloudiness | | | | |
| | Spring | -0.15 | 0.06 | 0.02 |
| | Autumn | 0.19 | 0.05 | 0.00 |
| | Late Morning | 0.04 | 0.04 | 0.28 |
| | Early Afternoon | 0.09 | 0.05 | 0.09 |
| | Late Afternoon | 0.02 | 0.03 | 0.43 |
| | Early Evening | 0.00 | 0.04 | 0.96 |
| Precipitation | | | | |
| | Spring | 0.07 | 0.07 | 0.33 |
| | Autumn | 0.07 | 0.09 | 0.42 |

*(Continued)*

**Table 3.** (Continued)

| Outcome | Predictors | Std. Estimate | SE | p |
|---|---|---|---|---|
| | Late Morning | 0.06 | 0.04 | 0.14 |
| | Early Afternoon | 0.09 | 0.03 | 0.01 |
| | Late Afternoon | 0.11 | 0.04 | 0.00 |
| | Early Evening | 0.07 | 0.02 | 0.00 |
| Sunshine | | | | |
| | Spring | 0.41 | 0.13 | 0.00 |
| | Autumn | -0.18 | 0.05 | 0.00 |
| | Late Morning | 0.05 | 0.05 | 0.33 |
| | Early Afternoon | 0.01 | 0.09 | 0.88 |
| | Late Afternoon | -0.10 | 0.08 | 0.22 |
| | Early Evening | -0.35 | 0.08 | 0.00 |
| Wind speed | | | | |
| | Spring | 0.09 | 0.03 | 0.01 |
| | Autumn | 0.07 | 0.03 | 0.02 |
| | Late Morning | 0.04 | 0.04 | 0.27 |
| | Early Afternoon | 0.02 | 0.04 | 0.59 |
| | Late Afternoon | -0.06 | 0.04 | 0.20 |
| | Early Evening | -0.06 | 0.06 | 0.33 |

*Note.* Sex coded as men = 0, and women = 1; Sunshine = Sunshine duration.

different occasions were below 30 Celsius degrees. We found that participant's baseline affect did not depend on any other conditions. Thus, we conclude that differences in season, weather, and time of day have little impact on baseline affect among participants for most laboratory research schedules. As our study had a reasonable sample size resulting in high statistical power, we believe that the null results are robust. Our work corresponds well with other large-scale integrative projects that indicated the non-significance of occasion-specific factors in their effect on research participants' characteristics [39].

Our findings support other research indicating that high ambient temperature is associated with lower positive affect [23, 32, 34]. Some studies suggested the opposite [27, 31], yet they did not consider the seasonal variations. To address the fact that weather is often nested in seasons, we built a multilevel model that accounted for more variance. If we did not include seasons in our analysis, we found a positive correlation between temperature and affect, which might suggest that individuals feel better on warmer days or in warmer seasons. If we we examined the association between temperature and affect within each season, we found that people felt better when it was cooler outside. This finding suggests that it is important to control for seasons when examining the association between weather and affect. However, the effect should be interpreted as small. Thus, temperature differences of as much as 30 Celsius degrees would not be likely to cause deviations from the affect among research participants of more than one standard deviation of the mean valence.

We found that the relationship between seasonal variation and affect was complex. First, individuals felt more positive affect in spring than in winter. Yet, at the same time, springs were much warmer than winters, and participants felt worse on days that were warmer. Consequently, these two effects operated together in opposing directions canceling each other out. This effect is puzzling because simple correlations indicated that the ambient temperature and spring (vs. winter) were positively related to affect. We suggest that the outcomes are best

interpreted as avoidance of thermal discomfort related to high temperatures in spring and autumn in our region and low temperatures in winter. Individuals might feel worse during spring and autumn heat, but they also might feel somewhat worse during the winter cold. Furthermore, our findings may suggest that other factors differentiate between the seasons that were not included in the analysis but might have influenced participants' affect.

Unexpectedly, we did not find influences of daily cycles on affect, when controlled for the weather conditions. Previous studies indicated that the circadian rhythm of affect was consistent with the standard work-rest pattern [11, 13]. In our study, participants could schedule the lab visit before, in between, after the work, due to their own preferences, which may indicate a non-standard work-rest pattern. Future studies could replicate our result with a more homogenous participants pool to account for the work-rest cycles.

## Limitations and futures directions

This study has several limitations. First, although we precisely match the weather conditions with the lab visit, there could be a difference between observed objective weather and the experienced weather, indicating the measurement error that could bias our findings [27]. For instance, individuals could differ in their exposure to current weather due to the chosen transportation, e.g., biking vs. arriving by car. Future studies could include time spent outside to control for exposure to the weather conditions.

Second, our data are restricted to observations from one country with a predominately continental climate. It is critical to repeat this analysis in countries with more extreme climates. For instance, individuals could experience the same temperature differently depending on what they are used to.

Third, we did not collect data in the summer, making it difficult to generalize our findings for the whole year. For humans, the most comfortable temperature is around 22˚C [64]. Thus the relation between weather and affect may be more explicit when the temperatures go beyond the comfort level.

Fourth, we controlled for only two individual characteristics, namely age, and sex. No other potentially useful moderating variables were assessed. Future studies could include individual differences that would moderate the association between weather and affect [23, 35] or time of the day and affect [13].

Fifth, as in previous studies, we found relatively small effect sizes, which were detectable due to our analyses' high power [23, 30, 35, 36]. However, as we pointed out, their statistical significance does not warrant predicting meaningful psychological differences.

Sixth, we accounted for several potential predictors in our model, with some of them intercorrelated. Thus, some interpretations of the parameters in our mediation model might be challenging or might be interpreted in different ways. For instance, it is not straightforward to conclude what is the meaning of season or time of day after controlling for the weather. We examined and ruled out the risk of multicollinearity, yet we cannot exclude the likelihood that some associations might have been spurious. This warrants further conceptual work and empirical studies that dissect several causal pathways initiated by one causal factor, i.e., different effects due to seasonal activities (e.g., duties, holidays, more time spent for outdoor leisure) or due to biological effects on the human body (e.g., thermal stress during spring or summer heats).

Finally, our findings have limited generalizability. Aiming to advance laboratory research practice, we focused on baseline affect measures among resting research participants in a well-controlled room environment. For instance, we aimed to keep 23 degrees Celsius temperature in the room, constant dim light, and external sound attenuation. Thus, these findings

generalize to individuals under specific conditions that isolate the room environment from the outdoor environment. The results might be different for other scenarios. For instance, individuals might be more prone to weather or time of day if they were less isolated from the outdoor environment, e.g., if the room temperature followed outdoor temperature or the intensity of ambient light (light intensity and light temperature) was influenced by outdoor light, or if participants might observe wind or rain through the window. Our null findings might not hold for laboratories that do not meet some of these standardization criteria, e.g., have poor air conditioning. Moreover, we tested affect among individuals who might have been focused on the upcoming research tasks, and consequently, defocused from other daily factors such as time, weather, or other seasonal activities. Thus, our findings are not generalized to other scenarios where individuals have less restricted focus and might be more attentive to external factors. Our findings also generalize to the effects of weather and time cycles on affect. Our approach does not rule out the possibility that weather and time cycles might affect other processes that are of interest to affective scientists, e.g., cardiovascular circadian rhythms [14, 65].

## Conclusions

This study provided novel evidence of how several external contextual factors influence baseline measurements of affect in laboratory studies. Despite an extensive scope of potential factors, we found that resting individuals, anticipating upcoming tasks, well-isolated from the outdoor, presented marginal affective propensity to weather and time-rhythm variation. This seems to suggest that as long as standardized room settings are kept constant, experimental research in affective science is robust to occasion-specific factors offering comparable levels of baseline affect among individuals who participate at a different time of year, time of day, or in different weather conditions.

## Supporting information

**S1 Table. Overview of studies characteristics.**
(DOCX)

**S1 File. Data for the study.**
(XLSX)

## Acknowledgments

We thank Jolanta Enko, Michał Misiak, and Michał Kosakowski, and other members of the Psychophysiological Laboratory at Faculty of Psychology and Cognitive Science at Adam Mickiewicz University for their contribution to funding acquisition and data collection in the initial experiments.

## Author Contributions

**Conceptualization:** Maciej Behnke, Lukasz D. Kaczmarek.

**Data curation:** Maciej Behnke, Magdalena Pietruch.

**Formal analysis:** Maciej Behnke.

**Funding acquisition:** Lukasz D. Kaczmarek.

**Investigation:** Maciej Behnke, Magdalena Pietruch.

**Methodology:** Maciej Behnke, Lukasz D. Kaczmarek.

**Project administration:** Maciej Behnke.

**Resources:** Maciej Behnke, Lukasz D. Kaczmarek.

**Software:** Lukasz D. Kaczmarek.

**Supervision:** Lukasz D. Kaczmarek.

**Validation:** Maciej Behnke.

**Visualization:** Maciej Behnke.

**Writing – original draft:** Maciej Behnke, Hannah Overbye, Magdalena Pietruch, Lukasz D. Kaczmarek.

**Writing – review & editing:** Maciej Behnke, Hannah Overbye, Magdalena Pietruch, Lukasz D. Kaczmarek.

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
