## [Decision Letter · Decision Letter 0]

17 May 2021

PONE-D-20-38859

How seasons and weather conditions influence baseline affective valence in laboratory research participants ?

PLOS ONE

Dear Dr. Behnke,

Thank you for submitting your manuscript to PLOS ONE. After careful consideration, we feel that it has merit but does not fully meet PLOS ONE’s publication criteria as it currently stands. Therefore, we invite you to submit a revised version of the manuscript that addresses the points raised during the review process.

Both Dr. Richard Lucas (who chose to reveal his identity as a reviewer) and I have carefully read your manuscript. On the whole, we both find the topic interesting, but see a number of significant concerns with the paper as written. As Dr. Lucas calls out, the sample, in and of itself, isn’t a problem, but there is insufficient information provided to allow us to evaluate if the different groups are matched on dimensions that may be relevant to this research. Put bluntly, could the results obtained be a result of the groups differing on some, currently, unreported dimension? The bigger problems, however, are analytical in nature. As Dr. Lucas points out, there are questions regarding the model selection, variable inclusion (multicollinearity), and causal inferences. I won’t repeat what Dr. Lucas called out, but I think his comments are completely on point and need to be addressed if this paper is to be published at PLOS ONE. Having said that, if the issues called out by Dr. Lucas can be rectified, I do believe this paper could add value to the scientific literature and I hope you take this opportunity to revise and improve your work.

We look forward to receiving your revised manuscript.

Kind regards,

Jeff Galak, PhD

Academic Editor

PLOS ONE

Journal Requirements:

2)  We note that the grant information you provided in the ‘Funding Information’ and ‘Financial Disclosure’ sections do not match.

3) Please ensure that you include a title page within your main document. You should list all authors and all affiliations as per our author instructions and clearly indicate the corresponding author.

Reviewers' comments:

Reviewer's Responses to Questions

**Comments to the Author**

1. Is the manuscript technically sound, and do the data support the conclusions?

Reviewer #1: Partly

2. Has the statistical analysis been performed appropriately and rigorously? 

Reviewer #1: No

3. Have the authors made all data underlying the findings in their manuscript fully available?

Reviewer #1: Yes

4. Is the manuscript presented in an intelligible fashion and written in standard English?

Reviewer #1: Yes

5. Review Comments to the Author

Reviewer #1: This paper uses data from approximately 1,000 participants to assess the associations between mood and weather, after adjusting for season and time of day. The question itself is interesting, and the data have some desirable features. They are not longitudinal, and the sample size might not be quite large enough to detect the types of weather effects that have been found in past studies; but the sample is okay. Below I describe some concerns and suggestions.

The overall sample comes from participants in seven different laboratory studies. The authors ignore this group structure, suggesting that participants from different studies do not differ from one another. However, no evidence for this is provided. I think it would be useful to provide more information about the different samples, including how many participated in different seasons, and whether there are any common variables that should not vary across seasons that could be used to show that these samples are indeed very similar. It might also be helpful to account for this group structure in the analysis itself, perhaps with a multilevel model.

It is not entirely clear exactly what model was tested in Figure 1. The text description implies that every path was included, in which case model fit would not be relevant, because the model would be saturated. Yet the authors do emphasize model fit (which happens to be just okay), which suggests that some paths were, in fact, omitted. So more detail about this model is needed.

I am very concerned about the authors' interpretations of the parameters from their mediation model. First, as I understand recent guidance from methodologists who focus on causal modeling, the association between predictors and outcome after controlling for mediators is difficult to interpret. This is true both for the interpretation of season effects after controlling for weather, but also for the interpretation of time of day effects controlling for weather (though these look consistent with the zero-order correlations).

More importantly, I worry that some of these associations may be spurious, perhaps because of multicolinearity. For instance, temperature is correlated with "spring" .81, and both are included in the model. Notably, the zero-order correlation between temperature and affect is positive, but very weak, which probably aligns well with theory and intuition. However, after controlling for the very strongly correlated "spring" variable, temperature now correlates negatively (and moderately) with affect, which doesn't make much sense theoretically or intuitively. So I'm very concerned that this is an artifact, and I think the authors need to do much more to ensure that that is not the case. For instance, if they simply looked at the association between temperature and affect in each season (perhaps after controlling for time of day), what do the associations look like? I appreciate the authors' goals of making sure that these contextual factors are addressed, but this does introduce some challenging analytic issues that need careful consideration and discussion. I do not have confidence from these analyses that temperature is associated with lower mood.

Related, the authors emphasize that "temperature partly mediated the effects of season on emotions" (p. 11), but they do not discuss the direction of this indirect effect, which is consistently negative. Spring and autumn are warmer than the winter and people are happier in the spring and autumn than the winter; but the indirect effect is actually negative, meaning that this indirect effect doesn't really "explain" the total effect in the way people expect. Rather the indirect effect "explains" why the total effect is not much higher. This should strange finding should not be glossed over, as I believe it would lead readers to (appropriately) question this result.

Minor:

In the very first paragraph, the authors state that "positive affect is elicited by favorable activities such as going on a trip with friends and playing in the park with a child, lying in a hammock overlooking the beach," citing a paper by Cohen et al. (2018). However, this sentence implies that actually being in these situations has been shown to be associated with increased positive affect, which the cited study does not show. Instead, the cited study provided participants with a list of situations and asked them to rate their *hypothetical* reactions to these events. This should be made clear, as—as currently written—the sentence implies that research shows that these affective reactions actually occur.

Similarly, when describing evidence for seasonality, the authors omit important features of the evidence that they review. Notably, at least some of the evidence they cite in support of the idea that those who do not suffer from seasonal affective disorder still experience lower moods in the winter do not actually study changes or even differences in moods across seasons. For instance, the Hardin et al. paper cited as evidence uses a retrospective questionnaire that asks participants to report whether their mood changes in the winter. Because of retrospection problems, this is not strong evidence, and these methodological features should be noted explicitly. Overall, the authors should provide more detail about the studies they review, because as currently written, the lack of detail can sometimes be misleading.

I understand that a figure with all paths would include lots of information. At the same time, readers sometimes skip to figures and tables, glossing over details about those tables and figures and text (which is why many style guides encourage authors to ensure that tables and figures stand alone). Because of this, I think that the figure, which excludes nonsignificant paths is somewhat misleading, as it doesn't reflect what model was actually tested. Given these concerns, combined with the fact that even nonsignificant paths can be important for interpreting overall results, I'd encourage the authors to consider some other way of describing these results; a full table is probably necessary, and not just in the supplemental material.

The authors use stars to indicate significance in Table 2, but they do not appear to be correct. For instance, there are no significant correlations in rows 9 through 14, even though many correlations in these rows exceed other significant correlations in other rows (including the .81 correlation between "spring" and "temperature." These should be checked and corrected.

----

I always sign my reviews: Rich Lucas

I also believe that the role of the reviewer is to identify strengths and weaknesses of a paper, not to provide a recommendation about acceptance versus rejection. Because editorial management systems require a response to questions about recommendations, I almost always select "revise and resubmit." This selection should not be interpreted as a recommendation, but rather as "I choose not to provide a recommendation."

6. PLOS authors have the option to publish the peer review history of their article (what does this mean?). If published, this will include your full peer review and any attached files.

Reviewer #1: **Yes: **Richard Lucas

---

## [Author Response · Author response to Decision Letter 0]

1 Jul 2021

Reviewer: 1

This paper uses data from approximately 1,000 participants to assess the associations between mood and weather, after adjusting for season and time of day. The question itself is interesting, and the data have some desirable features. They are not longitudinal, and the sample size might not be quite large enough to detect the types of weather effects that have been found in past studies; but the sample is okay. Below I describe some concerns and suggestions.

#1. The overall sample comes from participants in seven different laboratory studies. The authors ignore this group structure, suggesting that participants from different studies do not differ from one another. However, no evidence for this is provided. I think it would be useful to provide more information about the different samples, including how many participated in different seasons, 

and whether there are any common variables that should not vary across seasons that could be used to show that these samples are indeed very similar. It might also be helpful to account for this group structure in the analysis itself, perhaps with a multilevel model.

Reply: As suggested, we tested the two-level model, in which we nested individuals' data within experiments. We also added requested information about the number of participants tested each season and parts of the day within a specific study. Moreover, we also checked whether seasons differed in participants' age, sex, or BMI and found differences in age, sex, but not BMI. Post hoc tests showed that we tested more women in winter and autumn than in spring. Furthermore, we tested younger participants in spring than in winter and autumn. To account for these differences, we controlled for participants' age and sex in the analyses.

Changes in the manuscript: 

See Methods section, line number 184-186; 191-198.

See Results section, line number 208-216.

#2. It is not entirely clear exactly what model was tested in Figure 1. The text description implies that every path was included, in which case model fit would not be relevant, because the model would be saturated. Yet the authors do emphasize model fit (which happens to be just okay), which suggests that some paths were, in fact, omitted. So more detail about this model is needed.

Reply: We presented our conceptual model in the revised manuscript (Figure 1). In our analysis, we regressed the affective valence on the mediators (weather conditions) and independent variables (seasons, part of the day). Age and sex were introduced as covariates for valence. Thus, some paths were omitted. In our hypotheses testing, we focus on the significance of specific direct paths and indirect paths.

Changes in the manuscript: See Methods section, line number 203, and Fig 1.

#3. I am very concerned about the authors' interpretations of the parameters from their mediation model. First, as I understand recent guidance from methodologists who focus on causal modeling, the association between predictors and outcome after controlling for mediators is difficult to interpret. This is true both for the interpretation of season effects after controlling for weather, but also for the interpretation of time of day effects controlling for weather (though these look consistent with the zero-order correlations) More importantly, I worry that some of these associations may be spurious, perhaps because of multicolinearity. For instance, temperature is correlated with "spring" .81, and both are included in the model. Notably, the zero-order correlation between temperature and affect is positive, but very weak, which probably aligns well with theory and intuition. However, after controlling for the very strongly correlated "spring" variable, temperature now correlates negatively (and moderately) with affect, which doesn't make much sense theoretically or intuitively. So I'm very concerned that this is an artifact, and I think the authors need to do much more to ensure that that is not the case. For instance, if they simply looked at the association between temperature and affect in each season (perhaps after controlling for time of day), what do the associations look like? I appreciate the authors' goals of making sure that these contextual factors are addressed, but this does introduce some challenging analytic issues that need careful consideration and discussion. I do not have confidence from these analyses that temperature is associated with lower mood.

Reply: We agree that interpretation of the model is challenging. We believe that this reflects the challenge to understand the phenomenon where multiple contextual factors (often intercorrelated) determine how people finally feel while approaching an experiment. We also believe that presenting the full model helps in observing and addressing this complexity and boils it down to essential components. For instance, we show that of the several weather components, temperature holds as predictive of affect while other factors are best fixed as neutral. As suggested, we addressed multicollinearity and found that the predictors were within the recommended range of Tolerance> .02 and VIF < 5. (Dodge, 2008; Everitt &Skrondal, 2010; James et al., 2013; Vittinghoff et al., 2011). We addressed this problem complex models interpretation in the limitations section.

We also agree that our paper might benefit from more emphasis on interpretation challenges, including more analytical work. Thus, we took additional efforts to elucidate our findings further. First, we run analyses in which we tested the model for each season separately. We found that people felt better when it was cooler outside in spring β = -.27, 95% CI [-.40, -.14]. and in autumn β = -.12, 95% CI [-.24, -.01]. The relationship in winter was not significant β = -.005, 95% CI [-.09, .08]. This might suggest that the outcomes are best interpreted as avoidance of thermal discomfort related to high temperatures that are frequent in spring and autumn in our region. Furthermore, the overall positive relationship between affect and temperature might be explained by differences between seasons rather than cumulative effects of differences within seasons. This shows the advantage of accounting for two factors (seasons and ambient temperature) as these two seem to have opposing effects, i.e., individuals feel worse in response to winters' low temperatures, but they also feel worse in spring (and autumn) once it is hot outside. 

References:

Dodge, Y. (2008). The Concise Encyclopedia of Statistics. New York: Springer.

Everitt, B. S.; Skrondal, A. (2010), The Cambridge Dictionary of Statistics, Cambridge University Press.

James, G., Witten, D., Hastie, T., & Tibshirani, R. (2013). An introduction to statistical learning (Vol. 112, p. 18). New York: Springer.

Vittinghoff, E., Glidden, D. V., Shiboski, S. C., & McCulloch, C. E. (2011). Regression methods in biostatistics: linear, logistic, survival, and repeated measures models. New York: Springer.

Changes in the manuscript: 

See Methods section, line number 186-189.

See Results section, line number 216-217; 223-238.

See Discussion section, line number 326-335.

#4. Related, the authors emphasize that "temperature partly mediated the effects of season on emotions" (p. 11), but they do not discuss the direction of this indirect effect, which is consistently negative. Spring and autumn are warmer than the winter and people are happier in the spring and autumn than the winter; but the indirect effect is actually negative, meaning that this indirect effect doesn't really "explain" the total effect in the way people expect. Rather the indirect effect "explains" why the total effect is not much higher. This strange finding should not be glossed over, as I believe it would lead readers to (appropriately) question this result.

Reply: As explained in our response to the previous point, we think that this finding indicates that increasing the complexity of the model better describes the complexity of the phenomenon but at the same time complicates the model interpretation. We suggest that the outcomes are best interpreted as avoidance of thermal discomfort related to high temperatures that are frequent in spring and autumn in our region and low temperatures that occur in winter. Individuals might feel worse during spring and autumn heat, but they also might feel somewhat worse during the winter cold. We clarified this issue in the discussion. We presented the relationship between seasonal variation and affect as a complex phenomenon requirng investigations that account for seasons and temperatures.

Changes in the manuscript: See Discussion section, line number 274-297.

Minor issues

#5. In the very first paragraph, the authors state that "positive affect is elicited by favorable activities such as going on a trip with friends and playing in the park with a child, lying in a hammock overlooking the beach," citing a paper by Cohen et al. (2018). However, this sentence implies that actually being in these situations has been shown to be associated with increased positive affect, which the cited study does not show. Instead, the cited study provided participants with a list of situations and asked them to rate their *hypothetical* reactions to these events. This should be made clear, as—as currently written—the sentence implies that research shows that these affective reactions actually occur.

Reply: We clarified this issue, and we made it clear that people evaluate situations as being related to positive or negative affect.

Changes in the manuscript: See Introduction section, line number 39-43.

#6. Similarly, when describing evidence for seasonality, the authors omit important features of the evidence that they review. Notably, at least some of the evidence they cite in support of the idea that those who do not suffer from seasonal affective disorder still experience lower moods in the winter do not actually study changes or even differences in moods across seasons. For instance, the Hardin et al. paper cited as evidence uses a retrospective questionnaire that asks participants to report whether their mood changes in the winter. 

Because of retrospection problems, this is not strong evidence, and these methodological features should be noted explicitly. Overall, the authors should provide more detail about the studies they review, because as currently written, the lack of detail can sometimes be misleading.

Reply: We provided additional details about the methods used in the studies mentioned in the introduction.

Changes in the manuscript: See Introduction section, line number 54-59; 86-94.

#7. I understand that a figure with all paths would include lots of information. At the same time, readers sometimes skip to figures and tables, glossing over details about those tables and figures and text (which is why many style guides encourage authors to ensure that tables and figures stand alone). Because of this, I think that the figure, which excludes nonsignificant paths is somewhat misleading, as it doesn't reflect what model was actually tested. Given these concerns, combined with the fact that even nonsignificant paths can be important for interpreting overall results, I'd encourage the authors to consider some other way of describing these results; a full table is probably necessary, and not just in the supplemental material.

Reply: We moved the table with the detailed results of the tested model to the manuscript. We considered the figure with all paths. As presented below, the figure might be unclear to some readers. Thus we only included an additional description of Fig. 2 in the manuscript. 

We are ready to add a figure with the full model if still requested. 

Changes in the manuscript: See Table 3, line number 257, and Fig 2, line number 253.

#8. The authors use stars to indicate significance in Table 2, but they do not appear to be correct. For instance, there are no significant correlations in rows 9 through 14, even though many correlations in these rows exceed other significant correlations in other rows (including the .81 correlation between "spring" and "temperature." These should be checked and corrected.

Reply: We corrected the Table. Thank you!

Changes in the manuscript: See Table 2, line number 251.

---

## [Editor Report · Decision Letter 1]

9 Aug 2021

How seasons, weather, and part of day influence baseline affective valence in laboratory research participants ?

PONE-D-20-38859R1

Dear Dr. Behnke,

We’re pleased to inform you that your manuscript has been judged scientifically suitable for publication and will be formally accepted for publication once it meets all outstanding technical requirements.

Of note, the one reviewer asked to review this revision had technical difficulties with the Plos ONE system, but emailed me his decision privately. In short, he believed that the revision successfully responded to his concerns.

Kind regards,

Jeff Galak, PhD

Academic Editor

PLOS ONE
---

## [Editor Report · Acceptance letter]

11 Aug 2021

PONE-D-20-38859R1 

How seasons, weather, and part of day influence baseline affective valence in laboratory research participants? 

Dear Dr. Behnke:

I'm pleased to inform you that your manuscript has been deemed suitable for publication in PLOS ONE. Congratulations! Your manuscript is now with our production department. 

Kind regards, 

on behalf of

Dr. Jeff Galak 

Academic Editor

PLOS ONE